# Bio-Inspired Thermal Conductive Fibers by Boron Nitride Nanosheet/Boron Nitride Hybrid

**DOI:** 10.3390/ijms252011156

**Published:** 2024-10-17

**Authors:** Jiajing Zhang, Pingyuan Zhang, Chunhua Zhang, Jiahao Xu, Leyan Zhang, Liangjun Xia

**Affiliations:** 1State Key Laboratory of New Textile Materials and Advanced Processing Technologies, Wuhan Textile University, Wuhan 430200, China; zshin_jj@hotmail.com (J.Z.); 15987618029@163.com (P.Z.); 2215043052@mail.wtu.edu.cn (J.X.); leyan_zhang2001@163.com (L.Z.); liangjun_xia@wtu.edu.cn (L.X.); 2College of Textile Science and Engineering, Zhejiang Science and Technology University, Hangzhou 310018, China

**Keywords:** thermal interface materials, boron nitride nanosheet, hybrid structure, flexible fiber

## Abstract

With the innovation of modern electronics, heat dissipation in the devices faces several problems. In our work, boron nitride (BN) with good thermal conductivity (TC) was successfully fabricated by constructing the BN along the axial direction and the surface-grafted BN hybrid composite fibers via the wet-spinning and hot-pressing method. The unique inter-outer and inter-interconnected hybrid structure of composite fibers exhibited 176.47% thermal conductivity enhancement (TCE), which exhibits good TC, mechanical resistance, and chemical resistance. In addition, depending on the special structure of the composite fibers, it provides a new strategy for fabricating thermal interface materials in the electronic device.

## 1. Introduction

Because of the notable increase in device power, effective heat dissipation has become increasingly vital in modern electronics [1]. If the heat cannot be quickly evacuated, the accumulated heat would lead to high temperature, which has a significant impact on device performance and reliability [2]. The significant thermal interface resistance, resulting from the micro-void between the interface surfaces of heat-producing devices and heat sinks, significantly suppresses heat transmission. To minimize thermal contact resistance and eliminate these micro-voids, thermal interface materials (TIMs) are essential [3]. As the matrix of TIMs, polymer resin is frequently utilized due to its low density, high elasticity, and processability [4]. However, effective heat conduction is not possible due to the low thermal conductivity (TC) of pure polymer resin, where the polyurethane (PU) is approximately 0.018~0.024 W/(m·K) [5]. To expedite heat transmission from heat-producing components to heat sinks, high through-plane temperature coefficients of TIMs are thought to be crucial [6]. For electronic packages, insulating qualities are just as important as outstanding heat conductivity performance. Numerous techniques have been developed to align 1D or 2D thermal conductive fillers vertically to improve through-plane TC. These techniques include the use of 3D printing [7], cutting rolling [8], bidirectional freezing [9], and electric/magnetic field alignment [10]. While these techniques can significantly enhance through-plane TC, the large-scale manufacturing of composites is typically constrained by intricate procedures.

One practical way to improve through-plane TC is to hybridize fillers with diverse dimensions and forms by incorporating them into the polymer [11]. Fillers working together might lessen phonon scattering, aid form filler junctions, and boost filler packing density [12]. Since a single sheet-like structure obstructs itself in in-plane alignment, a combined arrangement of sheet-like fillers is shown to be useful in creating thermal conductive channels along the vertical direction. Hu et al. enhanced in-plane TC to 15.13 W/(m·K) with 50 wt% OH-BNNS by fabricating highly thermally conductive nanocellulose-based composite films via vacuum-assisted self-assembly based on a dual bio-inspired design [13]. Qu et al. constructed stem-like composites by the mechanical inducing self-assembly method through shape memory effects of poly (propylene carbonate) inspired by natural material [14]. The filler resin composite inspired by natural materials could effectively form the directional thermal conductivity path. Quan et al. adopted the layer-by-layer method to fabricate the graphene oxide (GO), polyetheramine (PEA), and carbon nanotubes (CNTs) on the surface of carbon fibers inspired by the fish scale hierarchy, and the composite fibers obtained 51.5% enhancement of TC [15]. Chen et al. fabricated the multi-signal self-sensing composite inspired by the perception of multiple functions of skin, which is expected to be widely used in the shells of smart equipment [16]. Compared with the common methods, the bio-inspired strategy could effectively widen functional application.

One-dimensional fillers with high axial TC can be added to polymer/hexagonal boron nitride (h-BN) composites to further improve their TC [17]. Furthermore, because of their large aspect ratio, 1D fillers are advantageous for building continuous thermally conductive networks [18]. Numerous hybrid fillers, including h-BN/carbon nanofiber [19], h-BN nanosheets/ZnO [20], and h-BN nanosheets/silver nanowires [21], have been described by researchers. Although they are randomly distributed, the 1D fillers and h-BN are typically well-overlapped to form interconnected networks. In actuality, larger TC values at lower filler concentrations are produced by encouraging the alignment of the hybrid fillers [22,23]. When building effective thermally conductive networks, carbon fiber (CF), with a high aspect ratio and a TC of up to 900 W/(m·K), is a good option [24]. However, because of the high aspect ratio and ultrahigh modulus of CF, the electrical insulation of composites decreases dramatically with increasing CF content [25]. Additionally, the processability and flexibility of the composites significantly worsen. Therefore, a crucial problem in creating polymer/h-BN composites with high thermal conductivity and electrical insulation is figuring out how to obtain a high alignment of h-BN with an appropriate loading of CF and so leverage the synergistic impact of the hybrid fillers.

In this work, we proposed a novel method that uses wet spinning, chemical grafting, and hot pressing to fabricate polyurethane composite fibers with enhanced TC. This method is based on the alignment of boron nitride (BN) and its modification with 3-(trimethoxy-silicyl) propyl methacrylate (TMSPMA), which is induced by stretching during the wet-spinning process. First, wet spinning was used to create the composites that contained aligned BN and PU. This process was brought about by stretching in the flow channel to form raw thermal conductivity fiber (RTF). Afterward, chemical-grafted stacking was employed to produce the boron nitride nanosheet (BNNS) coating the surface of the RTF. To construct a coating surface with a steadily aligned network of BNNS, the hot-pressing procedure was used to enhance the interface between the RTF and BNNS to construct the bio-inspired thermal conductive fiber (BTF). BN-filling builds the horizontally connected sheet structure inside the presence of PU, and the BNNS coating surface also constructs the horizontally connected structure in the BTF. This hybrid structure resulted in the inter-outer-connected thermal conductivity path, exhibiting a 176.47% increase in TC compared to pure PU. The heat dissipation test is also used to illustrate the possible use of bio-inspired thermal conductive fibers as TIMs.

## 2. Results and Discussion

### 2.1. Structural Characterizations of BN@TM and BNNS@TM

Herein, the BTF with the BN and BNNS hybrid structure was fabricated by wet spinning followed by hot pressing. The arrangement of the BN was regulated by the draft force during the wet-spinning process. Then, BNNS was introduced onto the surface of RTF by chemical grafting and hot pressing. Figure 1a,b show the fabrication process of the BTF. The RTF formed the dense structure through a nonsolvent-induced phase separation (NIPS). Then, the BNNS was chemically grafted onto the surface of RTF to form a bionic surface villus structure and increase the contact area with the environment, which is beneficial to the heat transformation. Further, the samples were put into the hot-press machine for a short time under high temperature (~100 °C) and pressure (~5 MPa) to increase the interaction between BNNS and RTF and drive the TMSPMA-modified BNNS (BNNS@TM), which tends to be horizontally arranged on the surface of RTF. Additionally, the obtained BTF exhibited great flexibility and stretchability, as illustrated in Figure 1c.

The BN was exfoliated into BNNS by ultrasonic exfoliation. The BNNS particles in the obtained solution were well dispersed and the Tyndall effect was shown in Figure 2a. Figure 2b,e show the SEM images of the BN and BNNS particles. As can be seen from Figure 2b, the surface of the BN is relatively smooth, and the side thickness of the BN is relatively thick, which exhibits an obvious multi-layer structure. It can be observed from Figure 2e that BNNS shows a slight increase in roughness and irregular edge structure, indicating the successful stripping of BN. The characteristic peaks of the original BN are 780 and 1380 cm^−1^, which are attributed to the bending and stretching vibrations of B-N [26,27], respectively (Figure 2c). For BNNS, there is a hydroxyl peak at 3300 cm^−1^, indicating that BN undergoes partial hydroxylation during the ultrasound process (Figure 2f). Meanwhile, BNNS can be observed to have a characteristic peak of B-N at 1393 and 793 cm^−1^, and the peak position is slightly shifted compared with BN. The peak intensity at 793 cm^−1^ is more significant than that at 1393 cm^−1^, which is mainly due to the successful stripping of BN. In addition, no other new characteristic peaks were observed in BNNS.

In Figure 2f,g, the characteristic diffraction peaks of BNNS at 26.7, 41.7, and 55.1° correspond to the crystal plane (002), (100), and (004), respectively [28,29]. Compared with the XRD pattern of BN, the characteristic diffraction peaks of BNNS and BN are the same, and there are no other new diffraction peaks. These results show that no other impurities are introduced in the process of BN stripping, and the crystal structure of BN is not damaged by this stripping method. The stripping effect of BN can be characterized by the lamellae exposure degree of this two-dimensional material, that is, the relative content of (004) crystal face. The intensity ratio of the (004) to (100) crystal face diffraction peaks of BN and BNNS are 0.32 and 0.44, respectively. The diffraction peak intensity of the (004) crystal plane of BNNS is higher than that of BN, which further indicates that BN is successfully stripped by this method. Therefore, BNNS was successfully exfoliated from the BN particles.

### 2.2. Morphology and Structure of Thermal Conductive Fibers

Figure 3a shows the FTIR spectra of RTF and BTF. It can be seen from the figure that the N-H stretching vibration peak exists at 3312 cm^−1^ of RTF and BTF. There are asymmetric and symmetrical stretching vibration peaks of -CH_2_- at 2939 and 2853 cm^−1^, vibration peaks of C=O at 1730 and 1700 cm^−1^, and characteristic peaks of C-O-C at 1075 cm^−1^ [5]. The characteristic infrared peaks of BN appear at 1372 and 765 cm^−1^. The main structure of BTF is consistent with that of RTF, indicating that the main chemical structure of the composites has not changed.

The characteristic diffraction peaks of RTF and BTF are 26.7, 41.7, and 55.1° (Figure 3b). The I_002_/I_100_ values of RTF and BTF are 7.96% and 12.98%, respectively (Figure 3c). The I_002_/I_100_ values of BTF are higher than those of RTF because the BNNS particles grafted on the surface of RTF are more inclined to be arranged horizontally along the fiber axis under the action of hot pressing [28,29]. Furthermore, the mechanical property of fibers was performed on the Instron machine in Figure 3d. As shown in Figure 3e,f, the stress and strain of the RTF composite fibers are 10.25 ± 1.36 MPa and 564.15% ± 56.52, respectively. The stress and strain of BTF are 10.16 ± 0.84 MPa and 529.07% ± 84.10%, respectively. As can be seen from the calculation in Figure 3e,f, the Young’s modulus of RTF and BTF is 74.38 ± 27.40 MPa and 91.22 ± 15.59 MPa respectively, and the Young’s modulus of BTF is 22.64% higher than that of RTF. This is due to the increase in rigid BNNS on the surface of BTF compared with that of RTF. Figure 3(g1) illustrates the stretching broken cross-section, and Figure 3(g2,g3) show the tensile cross-section topography of BTF. It can be observed that the tensile cross-section of the composite fiber is relatively rough, the connection between TMSPM-modified BN (BN@TM) and the PU matrix is relatively tight, and the aggregate size of BN@TM is also small, indicating the good interface adhesion between BN@TM and the PU matrix.

Figure 3(h1–h3) shows the brittle cross-section topography of BTF. Some BN@TM tends to be arranged along the axial direction of the fiber, as I_002_/I_100_ values mentioned. BTF exhibits good mechanical properties, and the along the axial direction, the structure could supply the heat flow channel in the direction of the through-plane. Thus, the BNNS@TM on the surface of BTF endows good mechanical properties and good interfacial adhesion.

To visually highlight the good thermally conductive properties of BTF with a hybrid structure, thermal conductivity enhancement (TCE) was adopted using Equation (1):(1)$$\mathrm{TCE}=\frac{{\mathrm{TC}}_{F}-{\mathrm{TC}}_{\mathrm{PU}}}{{\mathrm{TC}}_{\mathrm{PU}}}$$
where TC_F_ (TC_BTF_ and TC_RTF_) are the TC of BTF and RTF, respectively.

The TCE of BTF (0.47 W/(m·K)) reached 176.47% with the hybrid structure, exhibiting good TC shown in Figure 4a. Figure 4b,c show the surface morphology of RTF and BTF. It can be seen that the surface of RTF and BTF is relatively rough, which results from the large amounts of BNNS particles. Compared with RTF, the surface of the BTF composite fiber is relatively flat, which is due to the existence of more oriented BNNS on the surface. From the surface of BTF, it can also be observed that more BNNS are arranged parallel to the axis of the fiber, which is conducive to effective heat transfer, as shown in Figure 4(d2). More BN@TM accumulates around the micropores, providing a path for heat transfer, as shown in Figure 3(g3). At the same time, there is no obvious gap between BN@TM and the PU matrix, indicating that the interfacial interaction between BN@TM and the PU is good. A large number of BN@TM is distributed throughout the cross-section shown in Figure 3(h2,h3), which provides a good path for heat conduction, illustrated in orange in the SEM image. It was found that there were no obvious gaps between the outer layer and the core layer at the edge. Additionally, the longitudinal surface of the BTF can also be observed from the tensile cross-section of the fiber. BNNS and BN hybrid structures are horizontally arranged along the fiber axis, and the interfacial interaction between the fillers and PU is still good after stretching. Thus, there are different directions of heat flow transfer in the RTF, as shown in Figure 4(d1), which exhibit good TC and 97.08% TCE. The TC and mechanical properties of BTF were compared in Figure 4e [30,31,32,33,34]. It could be seen that the BTF was equipped with better TC and mechanical properties in our work. Owing to the surface coating BNNS@TM, it supplies more TC paths, and the synergistic effect of the inter-outer, or outer-outer hybrid results in the increase in TCE.

The RTF and BTF were woven into plain textiles to test the heat dissipation performance. An infrared thermal imager was used to collect thermal images with a heat source temperature of 80 °C from 0 to 60 s. It can be seen from Figure 5a that at 10 s, the surface temperatures of RTF and BTF were 61.7 and 64.4 °C, respectively. At 60 s, the surface temperatures of RTF and BTF were 76.7 and 79.7 °C, respectively. It shows that BTF has better TC.

Figure 5b shows the TC stability of RTF and BTF textiles during the bending cycle. T_0_ is the temperature at which the composite fiber is placed in the oven at 80 °C for 15 min and then taken out for 20 s, while T is the temperature at which the composite fiber is folded for 200, 400, 600, 800, and 1000 times, respectively, and then placed in the oven at 80 °C for 15 min and then taken out for 20 s. The RTF and BTF textiles can maintain good TC after 1000 bending cycles. Figure 5c shows the heat-dissipating temperature of RTF and BTF textiles after immersing in ethanol/water solvents for 48 h. The RTF and BTF textiles were removed from the ethanol/water solvents and dried at 80 °C. Then, the composite textiles are quickly transferred to the platform, and the thermal imaging images are collected at 0, 10, 20, 30, 60, 90, and 120 s. It can be seen from the images that the temperature of BTF textiles at 10, 20, and 30 s is 38.8, 29.8, and 27.6 °C, respectively, which is 9.3, 8.3, and 6.0 °C lower than that of RTF textiles. The heat dissipation performance of BTF textiles is better than that of RTF textiles after immersing in ethanol/water solvents.

Figure 4(d1,d2),e show the TC stability of BTF textiles under cyclic cooling or heating treatment. Among them, the cycle test method of high temperature to room temperature is as follows: the BTF is placed in the oven at 80 °C for 15 min, the fiber is taken out, and the temperature of the surface of the BTF is recorded at room temperature. After the textiles are left for 1 min, the temperature of the surface of the BTF is recorded, and the cycle was carried out 40 times. The cyclic test method of high temperature–room temperature was as follows: the composite textiles were placed in the refrigerator at 2 °C for 15 min, then the textile was removed, and the temperature of the textile’s surface was recorded at room temperature. After the textile was left for 1 min, the temperature of the surface of the BTF continued to be recorded, and the cycle was carried out 40 times. The high temperature and the temperature after standing for 1 min of BTF textiles are almost unchanged after 40 high-temperature cycles, indicating that the BTF has good heat absorption and heat dissipation capabilities. The surface temperature of BTF after 40 cycles of low temperature is similar to that after standing for 1 min, indicating that low temperature has no obvious effect on the TC of textiles. In short, BTF has good TC stability in the range of 2~80 °C.

## 3. Materials and Methods

### 3.1. Materials

PU was purchased from BASF Co., Ltd. (Ludwigshafen, Germany). BN was provided by Macklin Chemical Technology Co., Ltd. (Shanghai, China). TMSPMA was purchased from Aladdin (Shanghai, China). N, N-methylformamide (DMF), toluene (TO), and isopropyl alcohol (IPA) were purchased from Sinopharm Chemical Reagent Co., Ltd. (Shanghai, China). DI water (conductivity ≤ 16 MΩ·cm) was made in the laboratory.

### 3.2. Fabrication of Composite Fibers

Modification of BN: A total of 228 g IPA was mixed with 12 g DI, a certain amount of TMSPMA was added, and then the pH of solvents was adjusted to 4.0 by AcOH. Further, 5 g BN was added under magnetic stirring. After that, the solution is centrifuged with a high-speed centrifuge (5000 rpm), purified with IPA, and repeated many times. Finally, the TMSPMA-modified BN powder was obtained by drying at 110 °C to remove residual solution and named BN@TM.

Preparation of BN nanosheets (BNNS): The 5g BN powders were stripped in 240 g mixed solvent of IPA/DI (9:1 wt%) for 6 h to obtain stable dispersion. Then, 10 g TMSPMA and a certain amount of AcOH were added to the BNNS supernatant. Finally, a modified BNNS solution was obtained, named BNNS@TMS.

Preparation of bio-inspired thermal conductive fibers (BTF): The spinning solution was prepared by mixing BN@TM:PU (3:2) and binary solvent (DMF:TO = 1:1) together and mechanically stirred (520 rpm) at room temperature for 2 h. After that, the mixture solution was injected into the wet-spinning system shown in Figure 1a. After leaving the extrusion device, the spinning stock immediately enters the DI coagulation bath to promote fiber forming. At this time, the raw thermal conductive fiber (RTF) is obtained by 80 °C drying and hot-pressing (100 °C, 5 MPa, 3 min). Then, the RTF was put into BNNS@TMS for 4 h, and washed by IPA. After the hot-pressing process, the RTF was decorated with the BNNS to obtain bio-inspired thermal conductive fibers (BTF).

### 3.3. Characterization

A scanning electron microscope (SEM) (JEOL JSM-IT300LV, Tokyo, Japan) was used to characterize the morphologies of powders and composite fibers. The crystalline structure of BN, BNNS, and the composite fibers was recorded with an X-ray diffraction (XRD) instrument (Malvern PANalytical Empyrean, Almelo, The Netherlands). The mechanical properties were assessed by a universal material tester (Instron 5943, Norwood, MA, USA). The chemical structures of the BN, BNNS, and composite fibers were investigated using Fourier transform infrared (FTIR) spectroscopy (Thermo Fisher Scientific Nicolet iS20, Waltham, MA, USA). Thermogravimetric analysis (TGA) was performed on a HITACHI STA300 instrument (Ibaraki, Japan) at a heating rate of 10 °C/min from laboratory room temperature to 800 °C. The thermal conductivities of the composite fibers were measured using a thermal constant analyzer instrument (Hot Disk TPS 2200, Gothenburg, Sweden) according to ISO/CD 22007-2. The thermal energy storage and release properties of all samples were investigated using an infrared thermal imaging camera (FLIR T620, Portland, OR, USA).

## 4. Conclusions

We have effectively created a highly horizontally arranged network of bio-inspired BNNS coating on a BN@TM/PU composite by using wet-spinning, chemical grafting, and hot-pressing methods. The high degree of horizontal–directional alignment in/on the matrix was confirmed by the SEM and XRD. These hybrid inter/outer networks offered abundant and consecutive heat transport channels that greatly enhanced the TC of the composite, which can approach 0.47 W/(m·K), exhibiting a 176.47% TCE compared with the pure PU. In chemical and mechanical resistance cyclic simulation tests, the composites with such hybrid networks demonstrated heat dissipation capabilities as effective as TIMs, highlighting their potential use in high-performance electronic devices.

## Figures and Tables

**Figure 1 ijms-25-11156-f001:**
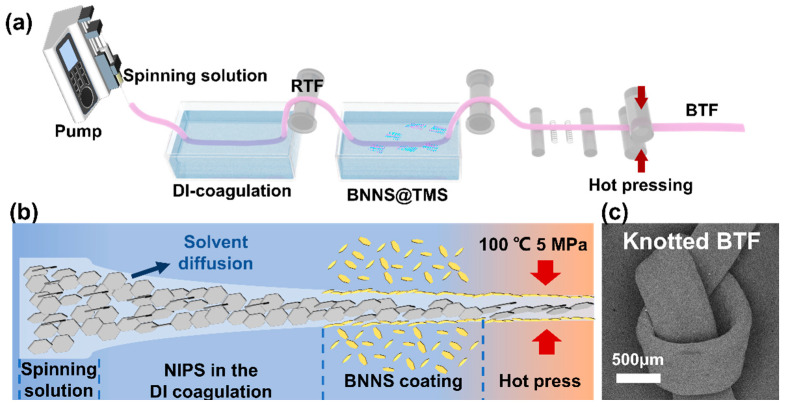
(**a**) An illustration for the fabrication of BTF; (**b**) schematic illustration of the BTF forming process; (**c**) SEM images of knotted BTF.

**Figure 2 ijms-25-11156-f002:**
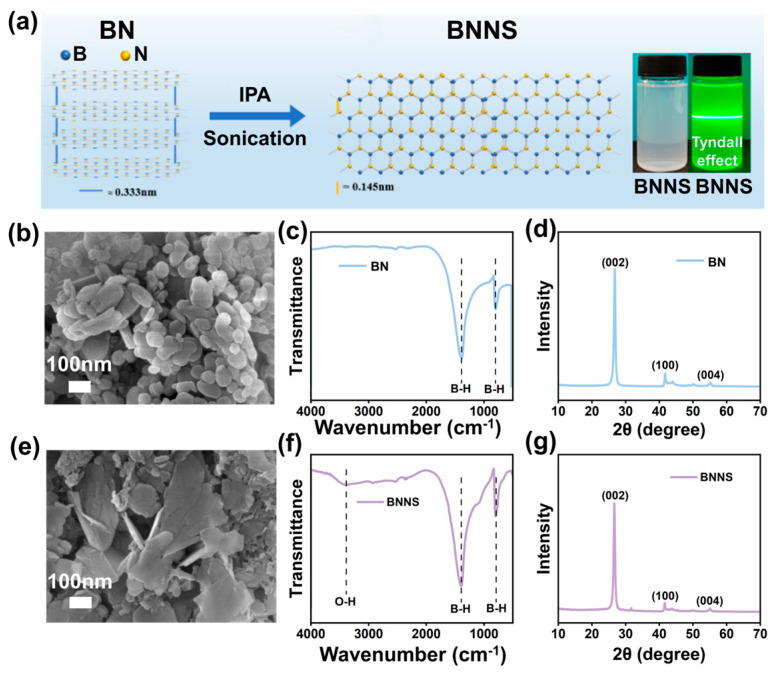
(**a**) The exfoliation process of BN; (**b**) SEM image, (**c**) FTIR spectra, and (**d**) XRD pattern of BN; (**e**) SEM image, (**f**) FTIR spectra, and (**g**) XRD pattern of BNNS.

**Figure 3 ijms-25-11156-f003:**
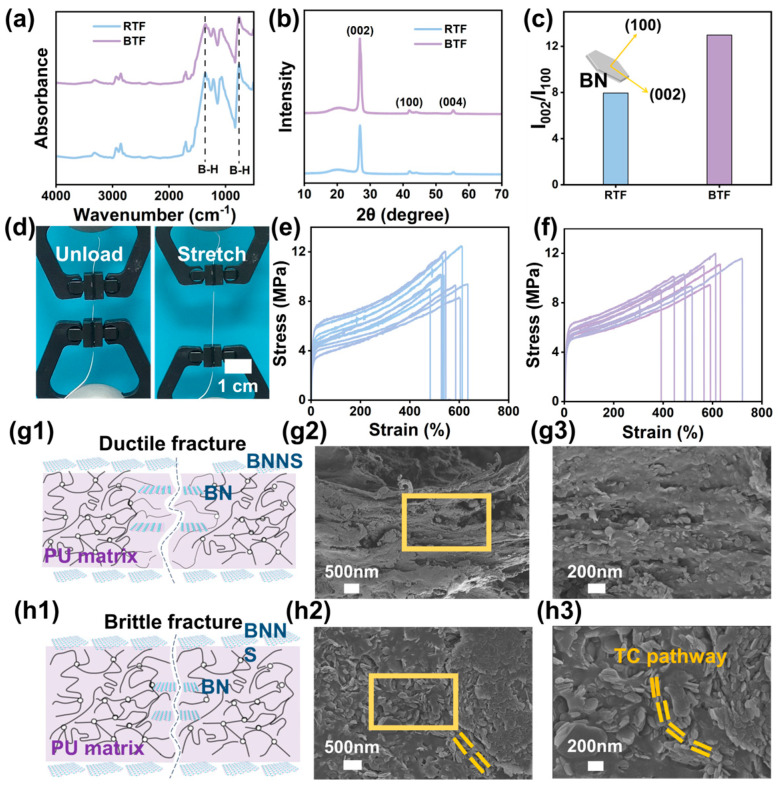
(**a**) FTIR spectra, (**b**) XRD patterns, and (**c**) the I_002_/I_100_ values of RTF and BTF; (**d**) photos of BTF before and after stretching; stress–strain curves of (**e**) RTF and (**f**) BTF. (**g1**) schematic illustration of mechanical broken BTF; mechanical broken cross-sectional SEM images of BTF (**g2**,**g3**); (**h1**) schematic illustration of a brittle-fractured BTF; (**h2**,**h3**) original cross-sectional SEM images of BTF.

**Figure 4 ijms-25-11156-f004:**
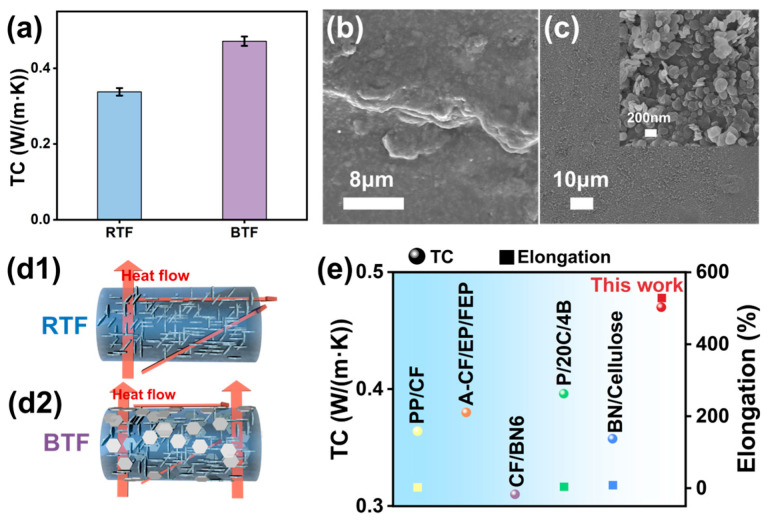
(**a**) TC of RTF and BTF; surface SEM images of (**b**) RTF and (**c**) BTF; the thermal transport mechanism of (**d1**) RTF and (**d2**) BTF; (**e**) TC and the mechanical property comparison of thermally conductive composites in the literature.

**Figure 5 ijms-25-11156-f005:**
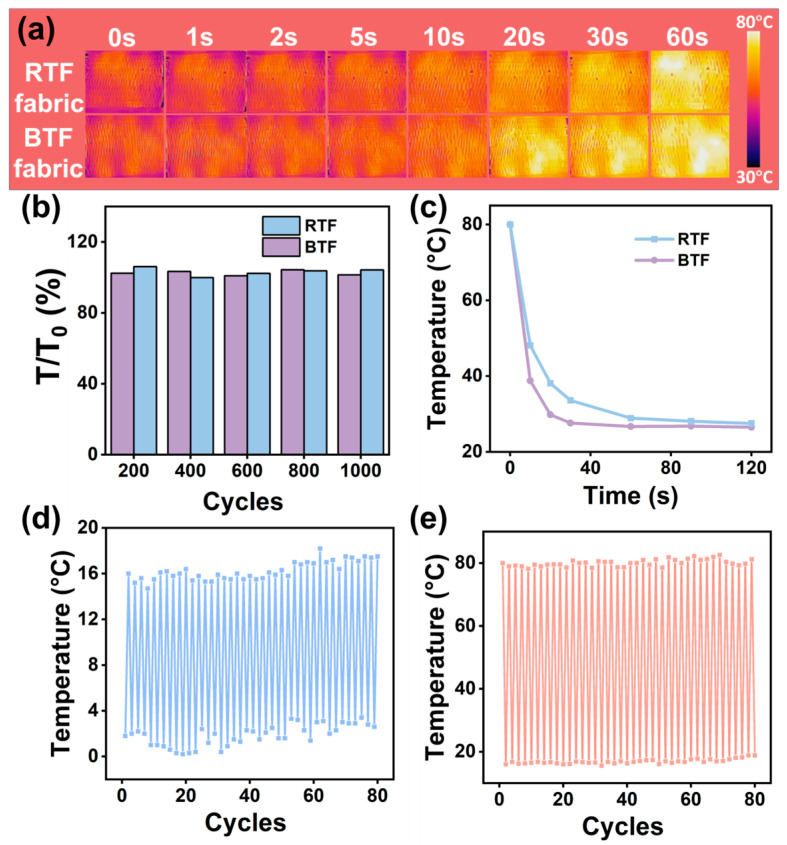
(**a**) Infrared thermal image of RTF and BTF textiles during the heating process (25 mm × 25 mm); (**b**) surface temperature of RTF and BTF textiles after different bending cycles; (**c**) surface temperature of RTF and BTF after 48 h ethanol/water immersion; surface temperature of BTF after cyclic (**d**) cooling and (**e**) heating.

## Data Availability

The raw data supporting the conclusions of this article will be made available by the authors on request.

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
