# Peer review of "Bio-Inspired Thermal Conductive Fibers by Boron Nitride Nanosheet/Boron Nitride Hybrid"

_ijms, 2024, doi:10.3390/ijms252011156_

Round 1
Reviewer 1 Report
Comments and Suggestions for Authors
The manuscript aims the creation of boron-nitride composites for thermal interface material.
There are lumped references which should be removed. All the reference should be assessed with at least half a sentence.
The structure should be changes. Results and discussion should follow the materials and methods.
The results and discussion should show the statistical evaluation of the samples, numerical values in tables or graphs, and the comparison to the existing similar material. Besides the other important mechanical characteristics should also be included.
Author Response
Response to Reviewer 1:
Comments for the Author
Comments:
The manuscript aims the creation of boron-nitride composites for thermal interface material.
- There are lumped references which should be removed. All the reference should be assessed with at least half a sentence.
Response:
We would like to thank the reviewer for the careful and thorough reading of this manuscript and for the thoughtful comments and constructive suggestions, which helped us to improve the quality of this manuscript. The lumped references have been removed, and more relevant literature has been added to the text, as shown below.
Page 2, “Quan et al adopted the layer-by-layer method to fabricate the graphene oxide (GO), polyetheramine (PEA), and carbon nanotubes (CNTs) on the surface of carbon fibers (CFs) inspired by the fish scale hierarchy. And the composite fibers obtained 51.5% enhancement of TC [15]. Chen et al fabricated the multi-signal self-sensing composite inspired by the perception of perception of multiple functions of skin which is expected to be widely used in the shell of smart equipment [16]. Compared with the common manufacture, the bio-inspired strategy could effectively widen the functional application.”
- The structure should be changes. Results and discussion should follow the materials and methods.
Response:
We are very grateful to you for the constructive advice. The structure of our manuscript followed the Template for IJMS downloaded from http://www.mdpi.com/files/word-templates/ijms-template.dot. In this template, the materials and methods follow the result and discussion.
Thanks again for your guidance on the manuscript!
- The results and discussion should show the statistical evaluation of the samples, numerical values in tables or graphs, and the comparison to the existing similar material. Besides the other important mechanical characteristics should also be included.
Response:
Thanks very much for your advice! The results and discussion have been shown by the numerical values in the manuscript. Indeed, the thermal conductivity and other mechanical properties of similar structure materials were explored and compared. Our work exhibited good thermal conductivity and mechanical properties.
Page 4, “The intensity ratio of the (004) to (100) crystal face diffraction peaks of BN and BNNS are 0.32 and 0.44, respectively. The diffraction peak intensity of the (004) crystal plane of BNNS is higher than that of BN, which further indicates that BN is successfully stripped by this method. Therefore, BNNS was successfully exfoliated from the BN particles.”
Page 5, “As shown in Figures 3e and 3f, the stress-strain curves of RTF and BTF exhibited the stress and strain of RTF composite fibers are 10.25 ± 1.36 MPa and 564.15% ± 56.52, respectively. The stress and strain of BTF are 10.16 ± 0.84 MPa and 529.07% ± 84.10%, respectively. As can be seen from the calculation in Figure 3e-f, the Young’s modulus of RTF and BTF is 74.38 ± 27.40 MPa and 91.22 ± 15.59 MPa respectively, and the Young’s modulus of BTF is 22.64% higher than that of RTF. This is because of the increase of BNNS on the surface of BTF compared with that of RTF due to the rigidity of BNNS.”
Page 6, “The TCE of BTF (0.47 W/(m·K)) reached 176.47% with the hybrid structure, exhibiting good TC shown in Figure 4a. Figures 4b and 4c show the surface morphology of RTF and BTF.”
Page 6, “The RTF and BTF were woven to 25 mm*25 mm plain textiles to test the heat dissipation performance. An infrared thermal imager was used to collect thermal images with a heat source temperature of 80 ˚C from 0 to 60 s. It can be seen from Figure 5a that at 10 s, the surface temperatures of RTF and BTF were 61.7 ˚C and 64.4 ˚C, respectively. At the 60 s, the surface temperatures of RTF and BTF were 76.7 ˚C and 79.7 ˚C, respectively. It shows that BTF has better TC.”
Reviewer 2 Report
Comments and Suggestions for Authors
The authors proposed a bio-inspired strategy for producing thermally conductive fibers made from boron nitride (BN) and polyurethane (PU). They specifically enhanced the thermal conductivity of the BN-PU composite fiber by applying an additional layer of BN on the exterior of the fiber. This approach resulted in a remarkable 176% increase in thermal conductivity compared to pure PU material. In this paper, the authors detail the fabrication process and provide comprehensive characterization of this fiber material. I recommend this paper for publication in the International Journal of Molecular Sciences after addressing the following comments:
- Please enhance the introduction by providing more information about thermal interface materials and recent developments in this field. Additionally, include a discussion on why the bio-inspired strategy was chosen to modify the BN-PU composite fiber.
- Some abbreviations used in this paper are presented in the wrong order. Please provide the full name of each term when it is first introduced in the text.
- Include a photograph of the sample fiber and the fabric used for thermal conductivity testing in Figure 5. It would also be beneficial to specify the dimensions of the sample used for this test.
- Please provide the thickness of the BN layer on the exterior of the fiber.
- Language polishing is needed throughout the manuscript.
Language polishing is needed.
Author Response
Response to Reviewer 2:
Comments for the Author
Comments:
The authors proposed a bio-inspired strategy for producing thermally conductive fibers made from boron nitride (BN) and polyurethane (PU). They specifically enhanced the thermal conductivity of the BN-PU composite fiber by applying an additional layer of BN on the exterior of the fiber. This approach resulted in a remarkable 176% increase in thermal conductivity compared to pure PU material. In this paper, the authors detail the fabrication process and provide comprehensive characterization of this fiber material. I recommend this paper for publication in the International Journal of Molecular Sciences after addressing the following comments:
- Please enhance the introduction by providing more information about thermal interface materials and recent developments in this field. Additionally, include a discussion on why the bio-inspired strategy was chosen to modify the BN-PU composite fiber.
Thanks very much for your advice. To satisfy the demand to fabricate components and devices that meet these requirements, new studies have gravitated towards biological systems found in nature to inspire the development of innovative lightweight engineered materials and structures (i.e., biomimicry). The complex lightweight architectures of these systems, consisting of organized structures at different length scales (i.e., hierarchical architecture), are the foundation for their incredible mechanical, hydrodynamic, optical, and conductive properties. Investigations into hierarchical architecture have facilitated the advancement of numerous bio-inspired engineered materials mimicking natural systems such as nacre, bone, plant leaves, and wood. As for the bio-inspired material, the surface became rougher which could make the contact area increase. The surface area of BTF increase could enhance the heat dissipation, which resulted in the enhancement of the thermal conductivity of BTF. Indeed, we summarized other thermally conductive composites, it could be seen that the BTF possessed good thermal conductivity and mechanical ductility in Appendix 1.
Appendix 1. Thermal conductivity and mechanical property comparison of thermally conductive composites in literature.
Page 2, “Since a single sheet-like structure obstructs itself in-plane alignment, a combined arrangement of sheet-like fillers is shown to be useful in creating thermal conductive channels along the vertical direction. Hu et al enhance in-plane TC to 15.13 W/(m·K) with 50 wt% OH-BNNS by fabricating highly thermally conductive nanocellulose-based composite films via vacuum-assisted self-assembly based on a dual bio-inspired design [13]. Qu et al construct stem-like composites by the mechanical inducing self-assembly method through shape memory effects of poly (propylene carbonate) (PPC) inspired by natural material [14]. The filler resin composite inspired by natural materials could effectively form the directional thermal conductivity path. Quan et al adopted the layer-by-layer method to fabricate the graphene oxide (GO), polyetheramine (PEA), and carbon nanotubes (CNTs) on the surface of carbon fibers inspired by the fish scale hierarchy. And the composite fibers obtained 51.5% enhancement of TC [15]. Chen et al fabricated the multi-signal self-sensing composite inspired by the perception of multiple functions of skin which is expected to be widely used in the shell of smart equipment [16]. Compared with the common methods, the bio-inspired strategy could effectively widen the functional application.”
Page 6-7, “Indeed, the TC and mechanical properties of this kind of composite fibers were compared in Figure 4e [30-34]. It could be seen that the BTF was equipped with better TC and mechanical property in our work.”
Figure 4. (a) TC of RTF and BTF; surface SEM images of (b) RTF and (c) BTF; the thermal transport mechanism of (d1) RTF and (d2) BTF; (e) TC and mechanical property comparison of thermally conductive composites in literature.
- 2. Some abbreviations used in this paper are presented in the wrong order. Please provide the full name of each term when it is first introduced in the text.
Response:
Thanks very much for your suggestions! We apologize for our mistake. The correct abbreviation presentation form is shown below.
Page 2, “This method is based on the alignment of boron nitride (BN) and its modification with 3-(trimethoxy-silicyl) propyl methacrylate (TMSPMA), ……Afterward, chemical-grafted stacking was employed to produce the boron nitride nanosheet (BNNS) coating surface of the RTF. To construct a coating surface with a steadily aligned network of BNNS, the hot-pressing procedure was used to enhance the interface between RTF and BNNS to construct the bio-inspired thermal conductive fiber (BTF)”
Page 3, “Further, the samples were put into the hot press machine for a short time under high temperature (~100 ˚C) and pressure (~5 MPa) to increase the interaction between BNNS and RTF and drive TMSPMA modified BNNS (BNNS@TM) tend to be horizontally arranged on the surface of BTF.”
Page 6, “the connection between TMSPMA-modified BN (BN@TM) and PU matrix is relatively tight, and the aggregate size of BN@TM is also small, indicating the good interface adhesion between BN@TM and PU matrix.”
- Include a photograph of the sample fiber and the fabric used for thermal conductivity testing in Figure 5. It would also be beneficial to specify the dimensions of the sample used for this test.
Response:
We are very grateful to you for the constructive guidance! The thermal conductivity of the composite fibers was tested by the transient Hot Disk instrument. The testing samples were required to be cut into the over 20 mm width and 200 μm thickness. Thus, it is hard to use fibers to test thermal conductivity and the composite fibers were woven into the plain textiles (25 mm*25 mm). And the corresponding description is below.
Page 7, “The RTF and BTF were woven into plain textiles to test the heat dissipation performance. An infrared thermal imager was used to collect thermal images with a heat source temperature of 80 ˚C from 0 to 60 s. It can be seen from Figure 5a that at 10 s, the surface temperatures of RTF and BTF were 61.7 ˚C and 64.4 ˚C, respectively. At the 60 s, the surface temperatures of RTF and BTF were 76.7 ˚C and 79.7 ˚C, respectively. It shows that BTF has better TC.”
.”
- Please provide the thickness of the BN layer on the exterior of the fiber.
Response:
We are very grateful to you for the constructive advice. BNNS were grafted onto the surface of RTF by chemical modification, in which the layers with more BN gathered in the interior and on the surface shown in Appendix Fig 1a. Further, to achieve good interface compatibility, we use hot pressing to make the BNNS horizontally arranged on the surface of the RTF fiber and embedded BNNS in the RTF fiber, so that the external BNNS and the internal BN are connected, which is conducive to the formation of the thermal conduction path. Therefore, no obvious surface and core structures were formed in the prepared BTF. However, the embedded structure of BNNS can be observed in the SEM shown in Appendix Fig 1b, resulting in a mixed structure of BNNS and BN, which is conducive to heat transfer from the inside to the outside.
Appenidx Fig 2 Surface SEM images of (a) RTF; (b) BTF.
- Language polishing is needed throughout the manuscript.
Response:
Thanks very much for your advice! Some grammatical mistakes have been amended.
Line 27,“resulting from micro-void between”has been correct to “resulting from the micro-void between”
Line 36, “For electronic packaging, ”has been correct to “ For electronic packaging,”
Line 44, “It is thought that fillers working together might lessen” has been correct to “ Fillers working together might lessen”
Line 100, “The RTF forms the tight” has been correct to “The RTF formed the dense ”
Line 119, “which is an obvious” has been correct to “which exhibits an obvious”
Line 165, “composites is not changed” has been correct to “composites has not changed”
Round 2
Reviewer 1 Report
Comments and Suggestions for Authors
The manuscript was improved, and all my comments were assessed. The manuscript should be accepted.